# Automatic uterus segmentation in transvaginal ultrasound using U-Net and nnU-Net

Dilara Tank [1,2]\*, Bianca G. S. Schor[3], Lisa M. Trommelen[1,4], Judith A. F. Huirne[1,4], Iacer Calixto[2], Robert A. de Leeuw[1,4]

**1** Department of Obstetrics and Gynecology, Amsterdam University Medical Center, location Vrije Universiteit Amsterdam, Amsterdam, North-Holland, the Netherlands, **2** Department of Medical Informatics, University of Amsterdam, Amsterdam, North-Holland, the Netherlands, **3** Department of Computer Science, University of Cambridge, Cambridge, United Kingdom, **4** Amsterdam Reproduction and Development, Amsterdam, the Netherlands

\* d.tank@amsterdamumc.nl

## Abstract

### Purpose

Transvaginal ultrasound (TVUS) is pivotal for diagnosing reproductive pathologies in individuals assigned female at birth, often serving as the primary imaging method for gynecologic evaluation. Despite recent advancements in AI-driven segmentation, its application to gynecological ultrasound still needs further attention. Our study aims to bridge this gap by training and evaluating two state-of-the-art deep learning (DL) segmentation models on TVUS data.

### Materials and methods

An experienced gynecological expert manually segmented the uterus in our TVUS dataset of 124 patients with adenomyosis, comprising still images (n = 122), video screenshots (n = 472), and 3D volume screenshots (n = 452). Two popular DL segmentation models, U-Net and nnU-Net, were trained on the entire dataset, and each imaging type was trained separately. Optimization for U-Net included varying batch size, image resolution, pre-processing, and augmentation. Model performance was measured using the Dice score (DSC).

### Results

U-Net and nnU-Net had good mean segmentation performances on the TVUS uterus segmentation dataset (0.75 to 0.97 DSC). We observed that training on specific imaging types (still images, video screenshots, 3D volume screenshots) tended to yield better segmentation performance than training on the complete dataset for both models. Furthermore, nnU-Net outperformed the U-Net across all imaging types.

**Data availability statement:** We do not have permission from the participants to share the dataset publicly online, as they did not give their consent. In addition to that we are not allowed to share the dataset because of Dutch privacy constraint laws. We do have permission to share the dataset with other researchers, provided their research question aligns with the signed patient information folder that the study participants provided. We can also only share a dataset when a data sharing agreement has been signed between the parties that want to share the data. This is, of course, something we facilitate if a party wishes to perform an IPD, for example. The decision to share data with other parties and to determine if this is in line with the signed informed consent is up to the principal investigator and the legal team of the Amsterdam UMC. This is a team decision. The AI lab at the Amsterdam UMC gynecology department (Div4-ailabgynaecology@amsterdamumc.nl) serves as the primary point of contact for other parties.

**Funding:** The author(s) received no specific funding for this work.

**Competing interests:** The authors have declared that no competing interests exist.

Lastly, we report the best results using the U-Net model with limited pre-processing and augmentations.

## Conclusions

TVUS datasets are well-suited for DL-based segmentation. nnU-Net training was faster and yielded higher segmentation performance; thus, it is recommended over manual U-Net tuning. We also recommend creating TVUS datasets that include only one imaging type and are as clutter-free as possible. The nnU-Net strongly benefited from being trained on 3D volume screenshots in our dataset, likely due to their lack of clutter. Further validation is needed to confirm the robustness of these models on TVUS datasets. Our code is available on https://github.com/dilaratank/UtiSeg.

---

## Introduction

Transvaginal ultrasound (TVUS) is an important imaging procedure for identifying pathologies related to the reproductive system in individuals assigned female at birth. TVUS scanning and interpretation can pose difficulties in pathologies like endometriosis [1] and adenomyosis [2], as both diagnosing processes require a high level of skill and experience to accurately identify and assess the extent of the diseases [1,2]. Studies have shown that the diagnostic delay in these disorders can be over seven years, and it's vital to decrease this delay [3]. One possible approach to address this is automated semantic medical image segmentation [4,5]. Semantic segmentation is the process of dividing an image into segments based on the type of structure they represent (e.g., abnormal and healthy tissue) [6]. This process provides a meaningful representation of images that can be conveniently analyzed and utilized [4] for early diagnosis [5]. Automated semantic medical image segmentation is achieved through deep learning (DL), which employs artificial neural networks with multiple layers to automatically extract representations of large amounts of data [7]. The potential of DL methods has been proven in gynecologic literature, for example, in the automated detection and classification of endometrial pathologies in patients with confirmed intrauterine lesions [8] and endometriosis classification [9]. In DL-based medical image segmentation, the U-Net model is currently the state-of-the-art [10] architecture, meaning that it is the most advanced and effective tool in this field [11]. Recent advances in DL-based segmentation have shown that nnU-Net, a self-configuring framework that uses U-Net as a backbone, can also annotate abnormalities in MRI scans with high accuracy [12,13]. Examples of successful image segmentation methods in ultrasound include prostate segmentation using a recurrent neural network and lymph segmentation using fully convolutional networks [14]. In the field of gynecology, semantic segmentation is less advanced, but progress has been made in areas such as ovarian cancer segmentation [15] and the segmentation of ovaries, follicles, and cysts [16]. To our knowledge, only one study [17] has tested an nnU-Net model to segment the uterine cavity using gynecological ultrasound 3D volumes. Overall, the literature about DL-based TVUS segmentation is rare, often case-based on

limited datasets, and unclear about dataset parameters [18]. Our study aims to address these limitations by training and evaluating the performance of two state-of-the-art DL segmentation methods on a TVUS dataset, including still, video, and 3D scans. We focus on uterus segmentation as a first step towards more complex TVUS segmentation tasks.

## Materials and methods

We describe our dataset preparation process and model architectures before outlining our training and validation stages and evaluation metrics.

### A. Dataset preparation

Because of the lack of publicly available TVUS segmentation datasets at the time of this study, we prepared a manually segmented dataset for uterus segmentation. This study reused data that was originally collected for a study on grading the sonographic severity of adenomyosis [19], as the transvaginal ultrasound images provided clear visualization of the uterus, making them suitable for our objectives. The data was collected in the gynecological outpatient clinic of the Amsterdam University Medical Center.

Six expert gynecologists examined adenomyosis patients using a Samsung WS80A or a Heral10 (Samsung, Seoul, South Korea) ultrasound machine with the V5-9 or the EV2-10A intra-cavity transducer. In 3D TVUS, a 90° angle in the Z-direction is used. Image resolution varies between 1194x701 and 1180x824 – as is common with ultrasound scans – according to machine variation. Informed consent was obtained from all participants in the study cited previously [19], which was ethically approved (2021.0035) and registered (NCT06117410). Ethical approval for this study was not required as no additional data collection was performed. From the original image bank, we include still images, video screenshots, and 3D screenshots — referred to as imaging types below. An example of the data is shown in Fig 1. The image bank was accessed in December 2023 for image collection. The authors could not identify individual participants during or after data collection. The image clutter, ultrasound machine imaging details, and metadata are purposefully left in to observe

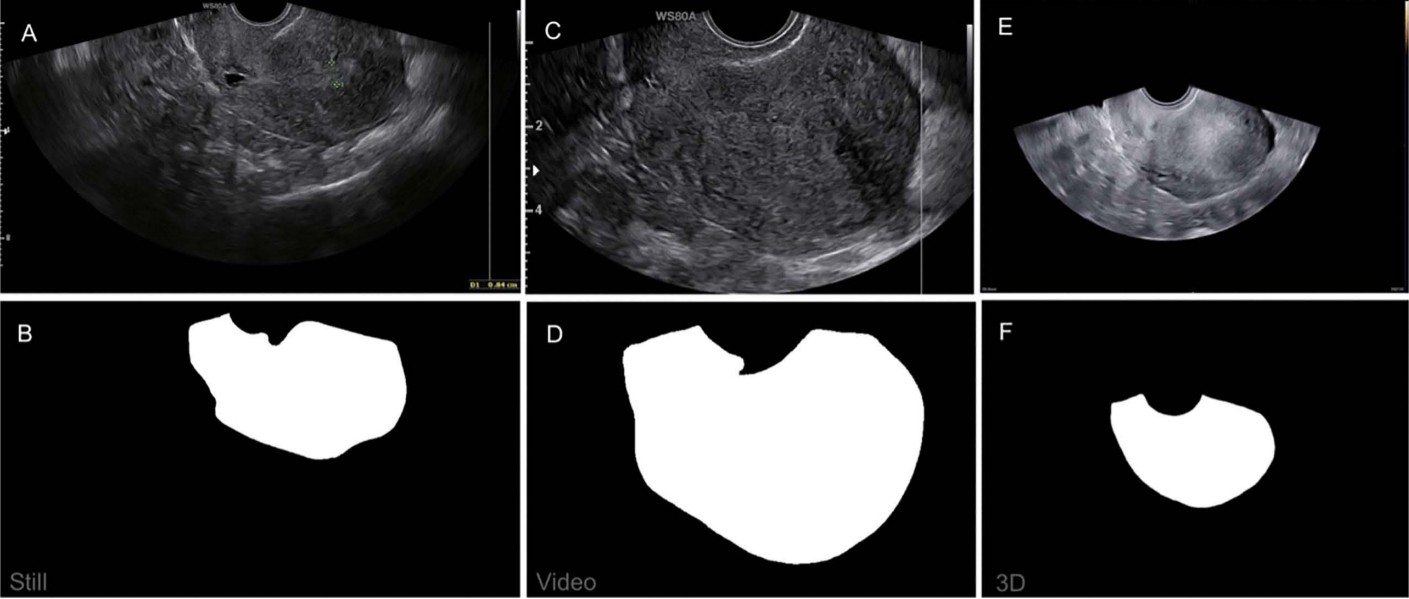

**Fig 1. Dataset example.** Still image (A) and mask **(B)**, video screenshot (C) and mask **(D)**, and 3D volume screenshot (E) and mask (F) from one patient.

their effect on the DL model performance. An expert gynecologist provided segmentation masks for all images using an iPad Pro 2020 as the "golden standard", usually called ground truth. Mask dimensions are the same as the original TVUS image dimensions and have values of 0 for background and 1 for the region of interest, e.g., the uterus. We do not consider the reliability of the ground truth here because the focus of this study is the feasibility of using TVUS images.

## B. Deep-learning models

We use two state-of-the-art DL segmentation models, U-Net [10] and nnU-Net [13], for their strong performance in biomedical image segmentation [11]. Additionally, using two models allowed for a performance comparison while keeping the analysis manageable.

The U-Net architecture is designed for biomedical image classification. It captures context and localization in images, making it possible to highlight regions of interest. While used in TVUS-related segmentation (e.g., cervical cancer [20] and ovarian cysts [21]), it hasn't been applied to 2D uterus segmentation. The 2D architecture used for this study is based on the original model by Ronneberger et al. [10]. Building upon the U-Net, Isensee et al. [13] proposed nnU-Net. This framework eliminates the manual iterative trial-and-error design process and automatically configures itself for any new task, including pre-processing, network architecture, training, and post-processing. To our knowledge, nnU-Net has only been utilized once in TVUS segmentation research [17]. These two architectures form the backbone of our approach to effectively segment 2D uterus images in TVUS data.

## C. Training stage

We train both U-Net and nnU-Net models four times: once on the full dataset and separately on each image type. This allowed us to assess whether combining imaging types improved model robustness and to evaluate the impact of each image type. To decrease the effect of a small dataset size, we use 5-fold cross-validation, where the dataset is split into five equal folds. The model is trained on four folds and tested on the fifth, rotating the test set with each iteration. This process ensures comprehensive training and evaluation across the entire dataset. For U-Net training, we follow a 70%, 10% and 20% dataset split for training, validation, and testing in each fold, consistent with the methodologies of Jin et al. [20] and Hu et al. [22].

Details for U-Net training are provided in Table 1. Hyper-parameter tuning is performed on the first fold, with optimized parameters for the remaining folds. We vary the batch size (4, 8, 16, and 32 images) and image resolutions (64x64,128x128, 256x256, and 512x512 pixels), to match the U-Net expected input shapes. We are aware that we are significantly downsizing the typically high-resolution TVUS scans. Lower resolutions, although reducing detail, help mitigate overfitting by focusing on essential features rather than specific details, encouraging the model to generalize better

**Table 1. U-Net training details.**

| Parameter | Description |
| --- | --- |
| Programming language and version | Python 3.9.18 |
| Framework | Pytorch Lightning 2.2.0.post0 |
| Optimizer | Adam |
| Learning rate | 0.001 with Stochastic weight averaging |
| Hardware | Nvidia A100-80 GPU |
| GPU memory | 40 GB (capped due to cluster-sharing policies) |
| Batch size (images) | 4 and 8 |
| Loss function | Dice loss |
| Epochs | Maximum of 150 with early stopping |

across different data. We also test pre-processing and augmentation approaches from previous TVUS segmentation research [22–24]. For pre-processing, we apply three techniques: we use image padding to standardize image dimensions while preserving shape; we apply Contrast Limited Adaptive Histogram Equalization (CLAHE) with a clip limit of 1 to enhance image contrast; and we use Gaussian blur with a kernel size of 5 to smooth the image and reduce noise. As an augmentation approach, we implement a 5-degree random rotation with a probability of 0.3, meaning that three images would be randomly rotated by five degrees for every ten images. This ensures the model handles small variations in rotation and better generalizes to new input data. The nnU-Net model, trained using version 2.0 of the framework from GitHub [25], utilizes resized images of 512x512 pixels.

### D. Model performance evaluation

Model performance is evaluated using the Sørensen-Dice similarity coefficient (also called Dice score) [26,27], a common evaluation metric for medical image segmentation [10,12,13,20,21]. It is calculated using the equation

$$\frac{2TP}{(TP + FP) * (TP + FN) + \partial}$$

The Dice score (DSC) measures the overlap between the ground truth (expert-annotated mask) and the predicted mask (true positives), accounting for false positives and negatives. This makes it a reliable metric for our analysis because it deals with size biases.

## Results

We successfully created a dataset of 1046 images and used it to train U-Net and nnU-Net models. We first report on our dataset features and model feasibility before going into our key findings on how to get the best model performance on our dataset.

### A. Dataset features

The data was collected at Amsterdam University Medical Centers (UMC), The Netherlands, between January 2021 and January 2024. An overview of the dataset features, along with details on the data collection process, can be found in Table 2. On average, the 124 patients included were 39 years old and between 23 and 55 years old. The study included

**Table 2. Dataset features.**

| Dataset Category | Details |
|---|---|
| Amount of Images<br>Total<br>Still images<br>Video screenshots<br>3D screenshots | <br>1046 (train = 716, test = 205, validation = 125)<br>122 (train = 86, test = 26, validation = 10)<br>472 (train = 331, test = 98, validation = 40)<br>452 (train = 326, test = 87, validation = 39) |
| Amount of patients<br>Average age<br>Age range<br>Sex | 124<br>39<br>23-55<br>Female |
| Place of data collection | Amsterdam UMC, the Netherlands |
| Time of data collection | January of 2021- January of 2024 |
| TVUS machine features | Samsung WS80A<br>Settings: Gain 44–71, Dynamic Range 100–123, Frequency 4.7 or 5.6 MHz.<br>Heral10 (Samsung, Seoul, South Korea)<br>Settings: Gain 86, Dynamic Range 50, Frequency 34–46 Hz. |
| Original image resolutions | Between 1194x701 and 1180x824 pixels |

consecutive premenopausal women with sonographic signs of adenomyosis. The final dataset we collected as part of this feasibility study consists of 1046 2D TVUS images, including 122 still images, 472 video screenshots, and 452 3D volume screenshots. The dataset composition is further highlighted in Table 2.

## B. Feasibility

The resulting evaluation scores for the U-Net and nnU-Net models on all four datasets show mean Dice scores ranging from 0.6528±0.27 to 0.9689±0.002 (see last row of Table 3). We performed manual pre-processing and hyperparameter optimization on the first fold for all U-Net training (see the first row of Table 3), which provided higher Dice scores than other folds. Training on specific imaging types (still images, video screenshots, 3D screenshots) tends to yield better segmentation performance than training on a full dataset (see Table 3). This is mainly observed in the U-Net results, where the Dice scores are consistently higher in the individual imaging types than in the full dataset. This trend also seems to be true for the nnU-Net, except for the still images, where the mean performance is lower than that of the full dataset. In the second and fifth folds, the optimal U-Net model shows outlier performance compared to other folds, with Dice scores around 0.30. Upon further examination, we observed that the training Dice scores for these folds were relatively high, around 0.80, and comparable to the training scores we observed for other folds. This discrepancy suggests that the outliers may be due to a significant difference between the test and training sets, causing poor generalization.

## C. Optimal U-Net training, augmentation, and pre-processing configuration

To optimize the performance of the U-Net training, we manually experiment with pre-processing, training, and augmentation configurations (see Table 4). We found that pre-processing and augmentation combinations do not yield the best results, and the U-Net model performance increases with configurations that use limited pre-processing and augmentations. The optimization is done only on the first fold of our datasets (see Table 3). As a reminder, the nnU-Net framework manages these configurations automatically, so we do not report on that.

Models trained on the full dataset, only still images, and only video screenshots have the best performance with an image resolution of 256x256 pixels. The model trained on the 3D screenshots performs best with an image resolution of 128x128 pixels. This might imply that the still images and video screenshots benefit from more detail and that the 3D screenshots provide sufficient contextual information, even at a lower resolution. All trained models seem to prefer a small batch size of eight for the models trained on the full dataset and the still images only, and a batch size of four images for

**Table 3. U-Net and NnU-Net 5-Fold Cross-Validation on TVUS uterus segmentation test set (Dice scores).**

| | | Full dataset | Still images | Video ss | 3D ss |
|---|---|---|---|---|---|
| 1st fold | U-Net | 0.8898 | 0.8981 | 0.9125 | 0.8966 |
| | nnU-Net | 0.8114 | 0.7206 | 0.8697 | 0.9716 |
| 2nd fold | U-Net | 0.3316 | 0.6709 | 0.6726 | 0.6862 |
| | nnU-Net | 0.8272 | 0.7035 | 0.8844 | 0.9692 |
| 3rd fold | U-Net | 0.8672 | 0.7557 | 0.7022 | 0.7074 |
| | nnU-Net | 0.8226 | 0.7423 | 0.8830 | 0.9662 |
| 4th fold | U-Net | 0.8518 | 0.6708 | 0.6580 | 0.6599 |
| | nnU-Net | 0.8288 | 0.7396 | 0.8784 | 0.9689 |
| 5th fold | U-net | 0.3236 | 0.8597 | 0.8712 | 0.8109 |
| | nnU-Net | 0.8351 | 0.7062 | 0.8735 | 0.9686 |
| All folds mean (Dsc), std | U-Net | **0.6528**±0.27 | **0.7710**\*±0.10 | **0.7633**±0.12 | **0.7522**±0.10 |
| | nnU-net | **0.8250**±0.01 | **0.7224**±0.02 | **0.8778**±0.01 | **0.9689**\*\*±0.002 |

\* Indicates best U-Net performance, \*\* Indicates best nnU-Net performance. "SS" stands for screenshot, "DSC" stands for Dice score

**Table 4. Optimal U-Net training, augmentation, and pre-processing configurations for each dataset.**

| | Full dataset | Still images | Video ss | 3D ss |
|---|---|---|---|---|
| Pre-processing/augmentation approach | None | None | CLAHE, padding | CLAHE |
| Resolution (pixels) | 256x256 | 256x256 | 256x256 | 128x128 |
| Batch size (images) | 8 | 8 | 4 | 4 |
| Results on 1st fold test set (Dsc) | 0.8898 | 0.8981 | 0.9125 | 0.8966 |

'SS' stands for screenshot, 'DSC' stands for Dice score.

the models trained on the video screenshots and 3D screenshots only. The resulting Dice scores from training on these configurations range from 0.8898 to 0.9125 and correspond to the Dice scores of the first fold in U-Net model training in Table 3.

### D. NnU-Net outperforms U-Net

nnU-Net consistently outperforms U-Net on all imaging types (see Table 3). The highest mean nnU-Net performance, with a Dice score of $0.9689 \pm 0.002$, is achieved on the 3D screenshots, while the highest mean U-Net performance, with a Dice score of $0.7710 \pm 0.10$, is achieved on the still images, as we saw in Section A. The 3D screenshots seem to have the least clutter across all imaging types, as shown in Fig 1. Another comparison between the nnU-Net and U-Net models is that the nnU-Net exhibits more stable performance across different folds and imaging types. At the same time, the U-Net shows significant variance in performance. This stability is evident in the lower standard deviation of the nnU-Net's results compared to the U-Net.

A selection of the best, median, and worst visual examples from the U-Net and nnU-Net evaluation is shown in Fig 2. Looking at the nnU-Net results, the best model prediction overlaps almost perfectly with the ground truth segmentation, the manual segmentation made by an expert gynecologist. The median model prediction, although there is a noticeable deviation from the ground truth, still captures the general shape and location of the structure. The results show that the anatomical boundaries in the lower part of the image were not clear enough for the model, leading to an error in the segmentation. The worst model prediction has a similar shape to the ground truth but misses a large chunk on the right side of the image. The U-Net model predictions are quite similar, except for the worst model prediction, where the model did not predict anything, resulting in a Dice score near zero. The U-Net prediction lines are more pixelated than the nnU-Net ones. This means that the prediction line boundaries are more grainy, probably due to the lower image resolution of the U-Net.

### Discussion

In this feasibility study, we create a TVUS uterus segmentation dataset of 1046 images to show that the segmentation process is feasible using a U-Net and nnU-Net model. We use a databank with adenomyosis patients, but focus on uterus segmentation. Uterus segmentation serves as a foundational step towards more complex segmentation tasks and can significantly aid gynecologists in the future [4,5,17]. We have found that our nnU-Net model with a self-configuring setup outperformed all other experimental setups, achieving a mean Dice score of $0.9689 \pm 0.002$. Our optimal U-Net model achieved a comparatively lower Dice score of $0.7710 \pm 0.10$ but was nevertheless able to demonstrate the feasibility of automated segmentation on TVUS data. In addition to this, three key findings have emerged related to maximizing segmentation performance.

Pre-processing and augmentation approaches are important in medical imaging because they enhance the quality and diversity of training data, leading to improved model robustness and performance. However, we found that our dataset is different, which could be due to inherent dataset characteristics, i.e., low image resolution. Our results illustrate the

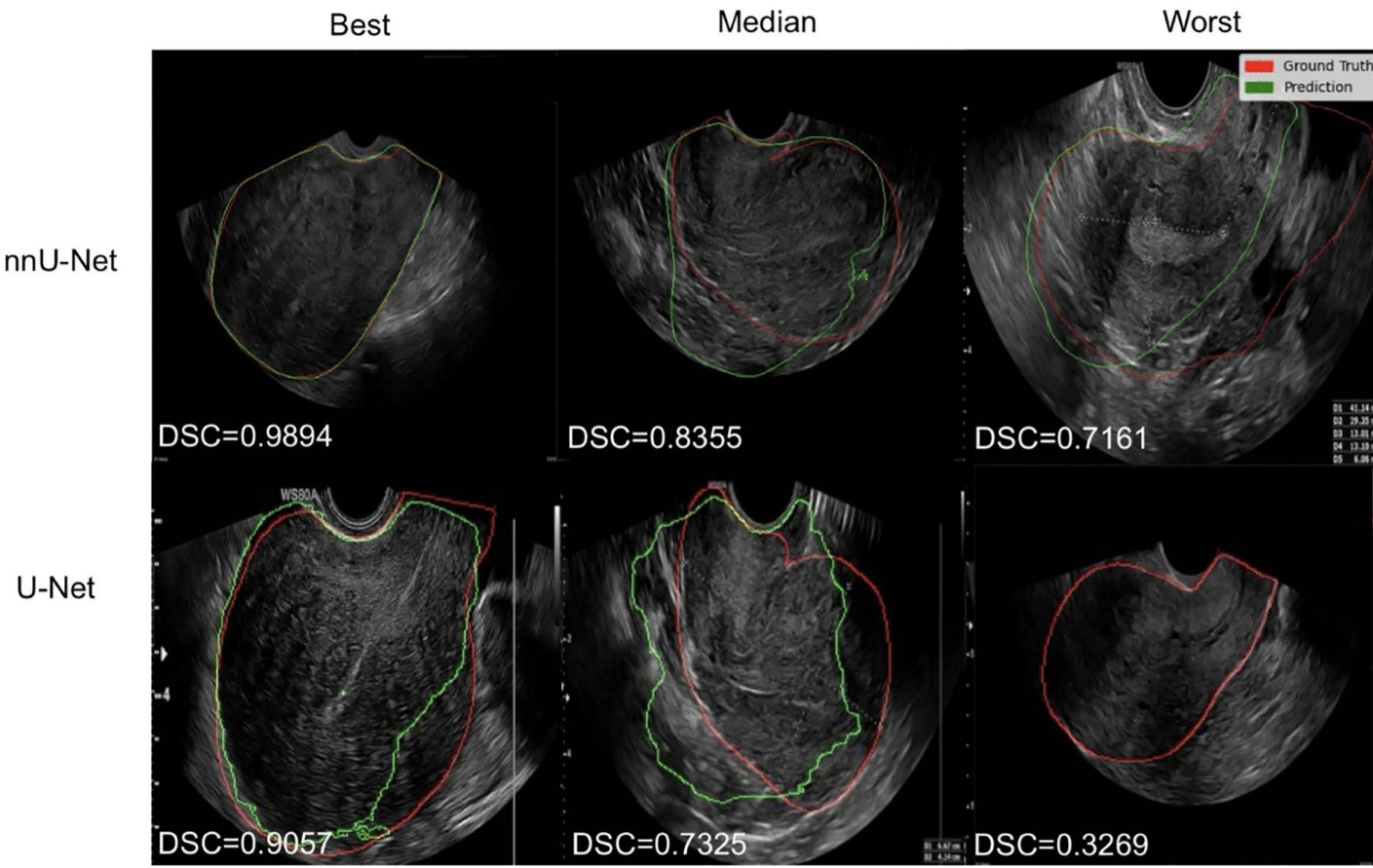

**Fig 2. Visual results picked from the best, median, and worst model folds.** TVUS scans of the uterus manually segmented by a doctor (red) and overlayed by nnU-Net and U-Net model predictions (green) and their corresponding Dice scores.

conclusions of a recent literature review [28], showing that in b-mode ultrasound segmentation, pre-processing and augmentation methods reach varying accuracy results depending on the quality of US images. Our results suggest that the data characteristics of our dataset, resulting in a lower quality of our images, indeed limit the benefits of these methods. Although the augmentation approaches did not yield a higher segmentation score, we still obtained strong results overall, so we do not consider this a limitation of our work.

Secondly, our results show that the self-configuring nnU-Net outperformed the manually configured U-Net. The resulting higher Dice scores and more stable performance can be attributed to the nnU-Net's ability to adapt more effectively to different data distributions and characteristics, reducing overfitting and enhancing model stability. Higher nnU-Net performance over U-Net performance is also demonstrated in a recent paper on segmenting critical anatomical structures in fetal four-chamber view images [29]. Because our pre-processing and augmentation methods were standard, we think that additional configuration of the U-Net model could potentially improve its performance for this task. However, this could risk overfitting the model to this task and making it less generalizable to larger or different datasets.

Lastly, we have found that training on specific imaging types yields better segmentation performance than training on a full dataset. This is because the full dataset, containing all three different imaging types, contains too much variability for the models to generalize well. The nnU-Net model achieves the highest performance on the 3D screenshots dataset, which could be explained by the fact that the 3D screenshots include the least amount of image clutter produced by

ultrasound machines. Our results confirm and reach similar Dice scores as the model performance reached in Boneš's et al.'s recent study [17]. Using 3D data in general has also been the advice of multiple TVUS segmentation papers [27–32]. This could have implications for collecting and creating future TVUS datasets. Compared to gynecological literature, e.g., ovarian cancer segmentation [15] and cyst and endometriosis segmentation [16], our results reach similar or higher Dice scores.

## A. Future perspectives

We now highlight future research directions. First, more research is needed to confirm how pre-processing and augmentation methods apply to other TVUS segmentation datasets. Second, for future TVUS segmentation research, we advocate for a standardized approach like nnU-Net, which ensures consistency and robustness across various datasets. Third, while further research is necessary, we recommend prioritizing collecting and using 3D screenshots for TVUS segmentation. Lastly, as a first step to using AI for uterine pathology, we advise a close collaboration between AI researchers and gynecologic pathologists, focusing on good-quality data collection and determining a reliable method to provide ground truth.

## B. Limitations

Given the challenges and costs associated with medical image acquisition and the need for publicly available TVUS segmentation datasets, the dataset size limits this feasibility study. However, our dataset is not smaller than most datasets in the field: for example, the works of Alwan et al., Marques et al., and Carvalho et al. [33–35] use datasets of fewer than 200 TVUS images. Another limitation is that our data is collected from a single center. To mitigate this to the best of our abilities, we tried to create variability in our data by including different imaging types in our dataset and data preprocessing and augmentation mechanisms. Lastly, the population used to train our models is not representative of all women, as our dataset included women with adenomyosis.

## C. Conclusion

This feasibility study shows that automated uterus segmentation on TVUS scans is feasible. Accurate TVUS scanning and interpretation can be challenging for gynecologists in complex pathologies. Proving the feasibility of automated uterus segmentation on TVUS scans is a step toward more complex automated segmentation tasks. The key benefit of these segmentation techniques lies in the ability to train, develop, and update a single AI system to support clinicians. Our results show that the nnU-Net could be a potential option for it. This provides a strong incentive to continue developing and eventually clinically evaluating TVUS segmentation methods.

## Author contributions

**Conceptualization:** Dilara Tank, Robert A. de Leeuw.

**Data curation:** Dilara Tank, Lisa M. Trommelen, Robert A. de Leeuw.

**Formal analysis:** Dilara Tank, Bianca G. S. Schor, Iacer Calixto, Robert A. de Leeuw.

**Investigation:** Dilara Tank, Bianca G. S. Schor, Robert A. de Leeuw.

**Methodology:** Dilara Tank, Bianca G. S. Schor, Iacer Calixto, Robert A. de Leeuw.

**Project administration:** Dilara Tank, Bianca G. S. Schor, Iacer Calixto.

**Resources:** Dilara Tank, Bianca G. S. Schor, Iacer Calixto, Robert A. de Leeuw.

**Supervision:** Bianca G. S. Schor, Iacer Calixto, Robert A. de Leeuw.

**Validation:** Dilara Tank, Bianca G. S. Schor.

**Visualization:** Dilara Tank.

**Writing – original draft:** Dilara Tank, Bianca G. S. Schor, Lisa M. Trommelen, Judith A. F. Huirne, Robert A. de Leeuw.

**Writing – review & editing:** Dilara Tank, Bianca G. S. Schor, Lisa M. Trommelen, Judith A. F. Huirne, Iacer Calixto, Robert A. de Leeuw.

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
