## [Decision Letter · Decision Letter 0]

8 Jul 2025

Dear Dr. Tank,

Thank you for submitting your manuscript to PLOS ONE. After careful consideration, we feel that it has merit but does not fully meet PLOS ONE’s publication criteria as it currently stands. Therefore, we invite you to submit a revised version of the manuscript that addresses the points raised during the review process.

We look forward to receiving your revised manuscript.

Kind regards,

Diego Raimondo

Academic Editor

PLOS ONE

3.  We noted in your submission details that a portion of your manuscript may have been presented or published elsewhere. [Yes, the underlying data we used was part of a study on adenomyosis

https://onlinelibrary.wiley.com/doi/full/10.1002/jum.16612] Please clarify whether this [conference proceeding or publication] was peer-reviewed and formally published. If this work was previously peer-reviewed and published, in the cover letter please provide the reason that this work does not constitute dual publication and should be included in the current manuscript.

4. In the online submission form, you indicated that [The data underlying the results presented in the study are available from Lisa Trommelen l.m.trommelen@amsterdamumc.nl].

Reviewers' comments:

Reviewer's Responses to Questions

**Comments to the Author**

1. Is the manuscript technically sound, and do the data support the conclusions?

Reviewer #1: Partly

Reviewer #2: Yes

2. Has the statistical analysis been performed appropriately and rigorously?

Reviewer #1: Yes

Reviewer #2: Yes

3. Have the authors made all data underlying the findings in their manuscript fully available?

Reviewer #1: No

Reviewer #2: Yes

4. Is the manuscript presented in an intelligible fashion and written in standard English?

Reviewer #1: Yes

Reviewer #2: Yes

Reviewer #1: In this study, the authors present a deep learning-based model for uterus segmentation in transvaginal ultrasound.

However, I believe the title of the manuscript should be reconsidered. Referring to the study as a demonstration of “feasibility” is questionable, given the substantial number of existing publications on ultrasound segmentation. The concept is no longer novel in itself.

There is also a conceptual inaccuracy in referring to both U-Net and nnU-Net as deep learning segmentation models. U-Net is a convolutional neural network (CNN) architecture, whereas nnU-Net is a self-configuring framework that uses U-Net as a backbone. This distinction is critical. Comparing the two is of limited value, as the superiority of nnU-Net as a comprehensive framework—including preprocessing, data augmentation, and parameter tuning—has been well established in the literature.

The introduction lacks coverage of key studies on deep learning-based segmentation in ultrasound beyond obstetrics. Important applications in breast, prostate, intraoperative brain tumor imaging, and thyroid ultrasound should be cited to better contextualize the work.

Furthermore, there is no clear description of the segmentation masks used—no details are provided regarding mask dimensions, range of values, or variability.

Lastly, and most importantly, we are currently facing a reproducibility crisis in the field of AI in medicine. It is imperative that studies of this nature be accompanied by a public repository including at least the trained model weights or checkpoints. This would enable other researchers to test the model on their own data and verify the findings.

Without addressing these issues, the manuscript falls short of the standards required for publication.

Reviewer #2: The aim of this study was to train and evaluate two deep learning (DL) segmentation models, U-Net and nnU-Net, to analyze the feasibility of DL-based uterus segmentation on transvaginal ultrasound (TVUS).

I have the following comments to the Authors:

• Please check for typos and tense mistakes throughout the text.

• Introduction

• Authors did not highlight enough the importance of the problem and why this study was necessary

• Authors should add in the introduction a brief revision about what is already known about the study subject, including other examples of DL applications in gynecologic pathologies [e.g. PMID: 38610993] and ultrasound diagnosis of adenomyosis [e.g. PMID: 38738458].

• Methods

• The study design is not clear, please state it explicitly at the beginning of materials and methods section.

• What was the setting in which the study was conducted?

• Authors should describe more in details the characteristics of the participants of the study used to create the image bank. In particular, how were selected the patients included in the study (inclusion and exclusion criteria) and how was selection bias avoided.

• Discussion

- Authors should include a brief qualitative resume of their results at the beginning of discussion

- Authors should include in the discussion a comparison of their study with similar studies already present in literature

- In order to make the article more praiseworthy, Authors should include a paragraph analyzing strengths and limitations of their study, for example discussing about the fact that the DL model was trained on a very specific population of uteruses (i.e. uteruses with adenomyosis).

**Do you want your identity to be public for this peer review?** For information about this choice, including consent withdrawal, please see our Privacy Policy

Reviewer #1: No

Reviewer #2: No

---

## [Author Response · Author response to Decision Letter 1]

12 Sep 2025

We would like to thank you for your constructive feedback. We have updated the paper accordingly and think the paper is much stronger as a result.

Journal requirements

1. Style requirements

After carefully reading the style requirements manual again, we have made appropriate changes.

2. Code sharing

We have included our GitHub repository and made mention of it in line 69-70 and 416.

3. Clarification on the part of the manuscript that has been published somewhere else

Our co-author of the paper, Lisa Trommelen, published a peer-reviewed paper called Grading Sonographic Severity of Adenomyosis

(https://onlinelibrary.wiley.com/doi/full/10.1002/jum.16612), where she aimed to develop a semi-quantifiable sonographic method to grade the severity of adenomyosis and assess the feasibility and interobserver reliability of this method. For this, she collected data, namely two-dimensional (2D) grayscale transvaginal ultrasound video clips and 3-dimensional (3D) volumes of the uterus of 35 women. This is the data that we then used in our study for another purpose: uterus segmentation. There is no risk of dual publication, as Lisa’s study focused on obtaining these data and analyzing them in a clinical setting, whereas our study focusses on creating automatic semantic segmentations of these images without a direct clinical conclusion or implementation. This is now explained in a clearer way in the subsection ‘dataset collection’.

4. Addressing the dataset availability

Because we are using a retrospective approach with data collected by a colleague for another project, we do not have ethical approval to make this data accessible online. We do appreciate this consideration and certainly we’ll implement it in our prospective work.

Reviewer comments

Reviewer 1

1. Reconsidering the title

We have now reconsidered the title to be: Automatic Uterus Segmentation in Transvaginal Ultrasound using U-Net and nnU-Net.

2. Distinguishing U-Net and nnU-Net more clearly

We aimed to keep the description accessible by not delving into the technical distinctions between U-Net and nnU-Net. However, we understand the importance of accuracy in terminology and have now changed it. In the introduction (line 86-87) we have now added a sentence for clarification.

3. Include more studies on deep learning-based segmentation in ultrasound beyond obstetrics & revision about what is known about the study subject (reviewer 2)

We appreciate this comment, and we have since incorporated the latest literature reviews in this field in the introduction, as well as studies from other fields. This answer also incorporates the feedback given by reviewer 2 in comment 3.

4. Segmentation masks

We have now clarified more on the segmentation masks in the dataset collection section.

Reviewer 1, comment 5: model weights

We have now added the u-net model weights in the public GitHub repo under the file ‘unetmodels.py’ along with instructions how to use them. The nnU-Net model weights were not saved during training.

Reviewer 2

1. Spelling and grammar mistakes

We have now fixed the additional spelling and grammar mistakes.

2. The importance of the problem and why this study was necessary

We have now changed the introduction to focus more on the importance of the problem and why our study was necessary, namely: automatic semantic segmentation has been proven to be very successful in medical imaging but in the field of gynecology, it is less advanced, especially when it comes to models trained on TVUS data. This is what our study aims to address.

3. Clarity on the study design

In the field of computer science and AI, there is no specific name for our study design. We take the following steps: we prepare our dataset, using the data collected by our co-author Lisa; we choose our model architecture; we train our model; and finally, we evaluate our model. We have now changed the materials and methods section to be clearer about the fact that we are not collecting data from patients as part of this study but merely using an already existing dataset to train our models.

4. The setting in which the study was collected

We have now added the sentence ‘The data was collected in the gynecological outpatient clinic of the Amsterdam University Medical Center.’ On line 129-130.

5. Patient characteristics and selection bias

As we were not designing the data collection, we put limited patient characteristics in this paper. However, some can be found in table 2. We have now also added the sentence ‘The study included consecutive premenopausal women with sonographic signs of adenomyosis.’ To clarify the inclusion criteria. After close inspection we didn’t identify any bias in our dataset, before moving on with the automatic analysis. We also didn’t notice any unusual results after the automatic segmentation.

6. A brief qualitative resume of the results

We have now changed our more quantitative summary to a qualitative one in the discussion.

7. Comparison to similar studies already present in literature

There are very few papers that are looking at different modalities of ultrasound scans, however, we have now included a sentence comparing our work to the recent literature reviews we have included in reviewer 1 comment 3.

8. Analyzing strengths and limitations of the study

We have now included a sentence that discusses the fact that our model was trained on a specific population of women in our limitations section, line 459-460.

---

## [Decision Letter · Decision Letter 1]

23 Oct 2025

Automatic uterus segmentation in transvaginal ultrasound using U-Net and nnU-Net

PONE-D-25-08028R1

Dear Dr. Tank,

We’re pleased to inform you that your manuscript has been judged scientifically suitable for publication and will be formally accepted for publication once it meets all outstanding technical requirements.

Kind regards,

Paolo Cazzaniga

Academic Editor

PLOS ONE

Additional Editor Comments (optional):

All reviewers' comments have been addressed by the authors, improving the overall quality of the manuscript, which can now be accepted for publication

Reviewers' comments:

Reviewer's Responses to Questions

**Comments to the Author**

Reviewer #1: All comments have been addressed

2. Is the manuscript technically sound, and do the data support the conclusions?

Reviewer #1: Yes

3. Has the statistical analysis been performed appropriately and rigorously?

Reviewer #1: Yes

4. Have the authors made all data underlying the findings in their manuscript fully available?

Reviewer #1: No

5. Is the manuscript presented in an intelligible fashion and written in standard English?

Reviewer #1: Yes

Reviewer #1: The authors have added clarifications and modifications to the manuscript, addressing my concerns and, in my opinion, improving the overall quality of the paper. Therefore, I recommend it for publication in its current form.

**Do you want your identity to be public for this peer review?** For information about this choice, including consent withdrawal, please see our Privacy Policy

Reviewer #1: No

---

## [Editor Report · Acceptance letter]

PONE-D-25-08028R1

PLOS ONE

Dear Dr. Tank,

I'm pleased to inform you that your manuscript has been deemed suitable for publication in PLOS ONE. Congratulations! Your manuscript is now being handed over to our production team.

Kind regards,

on behalf of

Dr. Paolo Cazzaniga

Academic Editor

PLOS ONE